# Vibration Control of a Wind-Excited Transmission Tower-Line System by Shape Memory Alloy Dampers

**DOI:** 10.3390/ma15051790

**Published:** 2022-02-27

**Authors:** Bo Chen, Xinxin Song, Wenbin Li, Jingbo Wu

**Affiliations:** 1Key Laboratory of Roadway Bridge and Structural Engineering, Wuhan University of Technology, Wuhan 430070, China; cebchen@whut.edu.cn (B.C.); wujingbo0618@whut.edu.cn (J.W.); 2Guangdong Power Grid Energy Development, Co., Ltd., Guangzhou 510160, China; cemqdwhut@163.com

**Keywords:** transmission tower, wind excitation, SMA damper, energy response

## Abstract

To be typical electrical power infrastructures, high-rise tower-line systems are widely constructed for power transmission. These flexible tower structures commonly possess small damping and may suffer strong vibrations during external excitations. The control approaches based on various devices have been developed to protect transmission towers against strong vibrations, damages, and even failure. However, studies on the vibrant control of wind-excited tower-line systems equipped with SMA dampers have not yet been reported. To this end, the control approach for wind-excited tower-line systems using SMA dampers is conducted. The mechanical model of the tower-line system is established using Lagrange’s equations by considering the dynamic interaction between transmission lines and towers. The vibration control method using SMA dampers for the tower-line coupled system is proposed. The control efficacy is verified in both the time domain and the frequency domain. Detailed parametric studies are conducted to examine the effects of physical parameters of SMA dampers on structural responses and hysteresis loops. In addition, the structural energy responses are computed to examine the control performance.

## 1. Introduction

High-rise truss towers, including television towers and transmission towers, are widely constructed for broadcast and electric energy supply. To be typical flexible structures, these truss towers commonly possess low damping and are prone to strong external excitations. If the load-induced strong vibration cannot be suppressed, possible damage and even failure are expected [1,2]. For example, a truss tower of more than 130 m in China collapsed when it was subjected to strong wind loadings [3]. The failure of transmission tower-line systems under strong earthquakes was also reported [4]. Therefore, many vibration control methods are developed to mitigate the excessive vibration of truss towers [5,6,7]. 

For television towers, vibration absorbers and dampers are firstly used for structural response control. The wind-excited television towers are protected by Yang et al. [8] and Wu et al. [9] by installing tuned mass dampers (TMD). In addition, the same devices are used in the vibration control of the 435 m Milad Tower in Tehran [10] and the 492 m World Financial Center Tower in Shanghai [11]. A similar wind-induced response control has been conducted using tuned liquid dampers [12,13]. The vibration control of television towers using dampers has also been performed in the past two decades [14,15,16]. Chen et al. [17] examined the seismic responses of a 340 m television tower reinforced using friction dampers. It was found that the implementation of friction dampers with optimal parameters can substantially reduce seismic responses. Zhang and Li [18] conducted seismic response control of a flexible truss tower using fluid dampers. They found that the vibration reduction effectiveness of fluid dampers was sensitive to the spectral characteristics of earthquake waves. For transmission towers, the response mitigation based on traditional control approaches was also performed. Chen et al. [19] examined the control performance of friction dampers on a wind-excited power transmission tower. The work on an example tower indicates that the application of friction dampers with optimal parameters could significantly reduce wind-induced responses of the transmission tower-line system for both the in-plane and out-of-plane vibration. In addition, they [20] also investigated the control efficacy of passive friction dampers on earthquake-disturbed transmission towers. They found that the best control performance of the transmission tower under a certain ground motion can be achieved only based on the optimal damper parameters and the control efficacy under different seismic excitations cannot keep optimal for all the time. The vibration control of a transmission tower under multi-component seismic excitations was conducted by Tian et al. [21] using TMD. Zhang et al. [22] applied a pounding TMD in a 55 m transmission tower for the seismic response control. Fluid dampers were accepted by Chen et al. [23] to reduce the impact responses of truss towers under cable rupture.

Various smart control devices, such as magnetorheological dampers, piezoelectric actuators, and shape memory alloy (SMA) devices, are recently gaining popularity in the vibration control of engineering structures [5,24,25]. The semi-active control of flexible truss towers under wind loading was carried out by Xu et al. [26] using piezoelectric friction dampers and by Chen et al. [27] using magnetorheological dampers. SMA is a typical smart material with many advantages including super-elasticity, fatigue resistance, and high strength [28,29]. Thus, SMA wires are widely used in vibration control to develop smart control devices, such as SMA dampers and SMA bracings [30,31,32,33]. Tian et al. [34] developed an SMA-based TMD for the seismic control of power transmission towers. Wu et al. [35] examined the seismic responses of a truss tower controlled by SMA dampers. 

However, studies on the vibrant control of wind-excited tower-line systems equipped with SMA dampers have not yet been reported. To this end, the new control method of transmission tower-line systems under wind excitations is proposed using SMA dampers. The mechanical model of a real tower-line coupled system was proposed using Lagrange’s equations by considering the dynamic interaction between transmission lines and towers. The vibration control method using SMA dampers for the tower-line coupled system was developed. The rational position of SMA dampers was determined by comparing three damper schemes. The feasibility of the proposed control method was verified through numerical analysis. Detailed parametric studies were conducted to investigate the effects of the physical parameters of SMA dampers on structural responses and hysteresis loops. Finally, the structural energy responses were computed to examine the control performance.

## 2. Model of a High-Rise Tower-Line Coupled System

### 2.1. Mathematical Model of Transmission Lines

The nonlinear equivalent method can be used to compute the dynamic responses of transmission lines using the Hamilton variational method and the Lagrange equation. As displayed in Figure 1, a transmission line is simulated by many masses and elements. Following the Hamilton variational principle, the line can be described as a series of generalized coordinates, namely the difference of the angle *θ* and element length *l*. If the line vibrates in the in-plane direction, the kinetic energy *T_line_* is [27]:(1)$$T_{line}=\sum_{i=1}^{4}\frac{1}{2}m_{i}({\dot{x}}_{i}^{2}+{\dot{y}}_{i}^{2})=T_{line}({\dot{\xi }}_{2},{\dot{\xi }}_{3},{\dot{\xi }}_{4},\delta {\dot{l}}_{1},\delta {\dot{l}}_{2},\delta {\dot{l}}_{3},\delta {\dot{l}}_{4},\delta {\dot{l}}_{5})$$
where *ξ**_i_* (*i* = 2,3,4) and *δ**_i_* (*i* = 1,2,3,4,5) are the structural generalized coordinates related to the *θ* and *l*, respectively [27]; *x_i_* and *y_i_* are the horizontal and vertical displacement of the *i*th mass, respectively. 

Similarly, structural potential energy *U_line_* is: (2)$$U_{line}=\sum_{i=1}^{4}m_{i}gy_{i}+\sum_{j=1}^{5}\frac{EA}{2}(\frac{{\left(l_{j}^{s}+\delta l_{j}\right)}^{2}}{l_{j}^{0}}-\frac{l_{j}^{s}{}^{2}}{l_{j}^{0}})$$
where *E* is Young’s modulus of the line; *A* is the cross-sectional area of the line; $l_{j}^{0}$ and $l_{j}^{s}$ are the initial and deformation length of the *j*th element. 

The equation of motion is derived based on Hamilton’s equation in terms of a set of generalized coordinates *q_i_*, namely *ξ* and *δ*:(3)$$\int_{t_{1}}^{t_{2}}\delta [T_{line}(t)-U_{line}(t)]dt+\int_{t_{1}}^{t_{2}}\delta W_{line}(t)dt=0$$
In which *W_line_*(*t*) denotes the virtual work.

Then, computing the variation yields into Lagrange’s equation:(4)$$\frac{d}{dt}\left(\frac{\partial T_{line}}{\partial {\dot{q}}_{i}}\right)-\frac{\partial T_{line}}{\partial q_{i}}+\frac{\partial U_{line}}{\partial q_{i}}=Q_{i}$$
where *Q_i_* is the generalized forcing function of the line.

If the line vibrates in the in-plane direction, the stiffness matrix $K_{l}^{in}$ is established by calculating the partial differential of the *U_line_* to the generalized displacement $\partial U/\partial \xi_{i}$ and $\partial U/\partial \delta_{i}$. Similarly, the mass matrix $M_{l}^{in}$ is established by calculating the partial differential of the *T_line_* to the generalized velocity $\partial T/\partial {\dot{\xi }}_{i}$ and $\partial T/\partial {\dot{\delta }}_{i}$. If the line vibrates in the out-of-plane direction, the line can be simulated as a suspended pendulum, as shown in Figure 1b. The system matrices **M***_l_*^*out*^ and **K***_l_*^*out*^ of the line are deduced similarly, as follows:(5)$$M_{l}^{out}=\left[\begin{array}{cc} m_{1} & \\ & m_{2} \end{array}\right]$$
(6)$$K_{l}^{out}=\left[\begin{array}{cc} \frac{m_{1}g}{l_{1}} & -\frac{m_{1}g}{l_{1}}\\ -\frac{m_{1}g}{l_{1}} & \frac{m_{1}g}{l_{1}}+\frac{(m_{1}+m_{2})g}{l_{2}} \end{array}\right]$$

### 2.2. Mathematical Model of Tower-Line Coupled System

The three-dimensional (3D) model of a real transmission tower in China is constructed using ANSYS, as shown in Figure 2a. If a 3D finite element (FE) model is used for the large-scale tower and is incorporated with SMA dampers subjected to wind excitations, the step-by-step dynamic computation will be unbearably time-consuming. This may make the parametric study tedious and impractical. In addition, the numerical simulation of wind loading of the 3D FE model is commonly carried out using the spectral representation method, which requires enormous series calculus. In practice, a lumped mass model is commonly adopted for vibration control and parametric studies, as shown in Figure 2b. 

A transmission tower-line coupled system is a complex continuous system consisting of many towers and lines. It is impossible and unnecessary to establish the system model considering all the lines and towers. Thus, the single tower associated with connected lines can be adopted in the dynamic analysis, as shown in Figure 3. The kinetic energy *T* and potential energy *U* of the tower-line system are expressed as follows:(7)$$T=T_{t}^{}+\sum_{j=1}^{nl}T_{l}^{\left(j\right)}$$
(8)$$U=U_{t}^{}+\sum_{j=1}^{nl}U_{l}^{\left(j\right)}$$
where *T_t_* and *U_t_* are the kinetic and potential energy of the single tower; $T_{l}^{\left(j\right)}$ and $U_{l}^{\left(j\right)}$ are the kinetic and potential energy of the jth line; *nl* is the number of all the lines. 

Then, Equations (7) and (8) can be substituted into the Lagrange equation. Similar to the computation of the transmission line expressed in Equation (4), the stiffness matrix of the entire coupled system $K_{}^{in}$ can be established in line with $\partial U/\partial \xi_{i}$ and $\partial U/\partial \delta_{i}$. The mass matrix $M_{}^{in}$ can be established in line with $\partial T/\partial {\dot{\xi }}_{i}$ and $\partial T/\partial {\dot{\delta }}_{i}$. For the out-of-plane vibration, the stiffness matrix $K_{}^{out}$ and mass matrix $M_{}^{out}$ of the entire coupled system are determined by combing the system matrices of the tower and lines. 

## 3. Mathematical Model of SMA Dampers

SMA wires have excellent inherited properties and can be used to fabricate smart energy-dissipating dampers. The SMA material can be described by the widely-used constitutive model [36,37,38]. The 2D and 3D configuration of an SMA damper is displayed in Figure 4. The SMA damper consists of the outer tube, inner tube, and circular plates. The SMA wires are incorporated in tension to dissipate energy during its reciprocal movement in vibration.

Figure 5 shows the hysteretic model of the widely-used Ni-Ti SMA material. In the figure, *M_f_* and *M_s_* are the martensite finish and start temperature, respectively. *A_f_* and *A_s_* are the austenite finish and start temperature, respectively; $\sigma_{M_{s}}$ and $\varepsilon_{M_{s}}$ are the critical stress and strain at martensite start temperature, respectively; $\sigma_{M_{f}}$ and $\varepsilon_{M_{f}}$ are the critical stress and strain at martensite finish temperature, respectively; $\sigma_{A_{s}}$ and $\varepsilon_{A_{s}}$ are the critical stress and strain at austenite start temperature, respectively; $\sigma_{A_{f}}$ and $\varepsilon_{A_{f}}$ are the critical stress and strain at austenite finish temperature, respectively; *ε_L_* is the maximum residual strain; *E_A_* and *E_M_* are Young’s moduli at the austenite and martensite phases, respectively. 

The relationships of strain and stress of the Ni-Ti SMA material shown in Figure 5 can be described based on different paths. 

The elastic stages (Paths O-A and E-O) are the full austenite stage. The damper force is given by:(9)$$u(t)=\frac{E_{A}A}{l_{w}}(d(t)-l_{w})$$
where *u*(*t*) is damper force; *d*(*t*) is the damper length with deformation; *A* is the cross-sectional area of a wire; *l_w_* is the original length of a wire.

The forward transformation stage (Path A-B) is the loading stage and the damper force *u*(*t*) is given by:(10)$$u(t)=\left[\sigma_{M_{s}}+\frac{\sigma_{M_{f}}-\sigma_{M_{s}}}{\varepsilon_{M_{f}}-\varepsilon_{M_{s}}}(\varepsilon (t)-\varepsilon_{M_{s}})\right]A$$

For the full martensite stage (Path B-C), the elastic deformation is observed and the damper force is:(11)$$u(t)=\frac{E_{M}A}{l_{w}}(d(t)-l_{w})$$

For the full martensite stage (Path B-D), the unloading process is observed and the damper force is:(12)$$u(t)=\sigma_{M_{f}}A+E_{M}\left[(d(t)-l_{w})-\varepsilon_{M_{f}}\right]A$$

The reverse transformation stage (Path D-E) is the unloading stage and the damper force *u*(*t*) is given by:(13)$$u(t)=\left[\sigma_{M_{s}}+\frac{\sigma_{M_{s}}-\sigma_{A_{f}}}{\varepsilon_{A_{s}}-\varepsilon_{A_{f}}}(\varepsilon (t)-\varepsilon_{A_{s}})\right]A$$
in which *ε*(*t*) is the stress of a wire.

## 4. Equation of Motion of the Controlled Structure

The equation of motion of the tower-line system equipped with SMA dampers is:(14)$$M\ddot{x}(t)+C\dot{x}(t)+Kx(t)=W(t)+Hu(t)$$
(15)$$M=\left[\begin{array}{cc} M^{in} & 0\\ 0 & M^{out} \end{array}\right]$$
(16)$$C=\left[\begin{array}{cc} C^{in} & 0\\ 0 & C^{out} \end{array}\right]$$
(17)$$K=\left[\begin{array}{cc} K^{in} & 0\\ 0 & K^{out} \end{array}\right]$$
(18)$$W(t)=\left[\begin{array}{cc} W^{in}(t); & W^{out}(t) \end{array}\right]$$
(19)$$u(t)={\left[\begin{array}{cccc} u_{1} & u_{2} & \cdots & u_{n} \end{array}\right]}^{T}$$
where x(t), $\dot{x}(t)$, and $\ddot{x}(t)$ are the displacement, velocity, and acceleration of the tower-line system, respectively; **M**, and **K** are the mass and stiffness matrices of the system, respectively; **C** is the Rayleigh damping matrix; **W**(*t*) is the wind-loading vector; **W***^in^*(*t*) and **W***^out^*(*t*) are the wind-loading vectors in the in-plane and out-of-plane direction, respectively; **u**(*t*) is the control force vector; **H** is the position matrix of **u**(*t*); *n* is the damper number. 

The wind excitations acting on the structural system are simulated using the spectral representation method. The vibration of a tower-line coupled system can also be illustrated using energy responses. The energy equations of the entire coupled system without and with SMA dampers are formed by integrating Equation (15). The total input energy from wind loading to the structural system *E_W_* is the sum of the kinetic energy *E_K_*, the strain energy *E_S_*, the energy dissipated by structural damping *E_D_*, and the energy dissipated by SMA dampers *E_C_*. 

## 5. Case Study

### 5.1. Structural and Damper Parameters

A 110 m transmission tower is displayed in Figure 2 and the span of the transmission lines is 800 m. Six platforms are constructed in the tower body and a horizontal cross arm is placed on the top to connect transmission lines. The tower members are fabricated by Q235 steel, which is a typical type of ordinary carbon structural steel in China. Q represents the yield limit of this material. The following 235 refers to the yield value, which is about 235 MPa. The Q235 steel is widely used in civil engineering structures because of the moderate carbon content and good comprehensive properties, such as strength, plasticity, and welding. The chemical composition of Q235 steel includes C, Mn, Si, S, and P. According to the contents of the different chemical compositions, the Q235 steel can be divided into four categories, A, B, C, and D. The chemical composition of Q235 steel is listed in Table 1.

The axial stiffness EA of the transmission line is 4.88 × 10^4^ kN. The weight per meter of the line is 1.43 kN/m. There are 1452 spatial beam elements and 353 nodes in the 3 D FE model of the example tower. The simplified dynamic model is established using a developed MATLAB program. The fundamental frequency of the single tower in the in-plane direction is 0.649 Hz and the counterpart in the out-of-plane direction is 0.643 Hz. The equation of motion is established using Rayleigh damping and solved using the Newmark-*β* method with a time interval of 0.02 s. The damping ratios of two fundamental frequencies are set as 0.01. 

Eight SMA dampers are evenly distributed in the tower body, as shown in Figure 6. Four dampers are installed in the in-plane direction (No. 1–4) and the other four are equipped in the out-of-plane direction (No. 5–8). The Young’s modulus of the SMA damper brace is 2.3 × 10^11^ N/m, and the cross-sectional area is 50 cm^2^. Considering the configuration of the tower, an SMA damper with an axial brace can be connected to a structural member in parallel, as shown in Figure 6. The control forces provided by the SMA damper directly act on the joint connection in the member’s axial direction. The material parameters of SMA dampers are as follows: the *M_f_* and *M_s_* of SMA materials are −46 °C and −37.4 °C, respectively; the *A_f_* and *A_s_* of SMA materials are −6 °C and −18.5 °C, respectively; the *C_M_* and *C_A_* of SMA materials are 10 MPa/°C and 15.8 MPa/°C, respectively; the *D_A_* and *D_M_* of SMA materials are 75000 MPa and 29300 MPa, respectively; the maximum residual strain *ε_L_* is 0.079. 

### 5.2. Peak Response Comparison

The vibration reduction factor (VRF) is adopted to assess the damper performance:(20)$$VRF=\frac{X_{n}-X_{c}}{X_{n}}$$
where *X_n_* and *X_c_* are the peak response without and with control, respectively.

Three damper location schemes are taken into consideration to compare the effects of damper position on control efficacy. For scheme 1, eight SMA dampers are installed on top of the tower body. Four dampers are installed in the in-plane direction with two dampers on the fifth floor and the other two on the sixth floor. Similarly, four dampers are installed in the out-of-plane direction with two dampers on the fifth floor and the other two on the sixth floor. For scheme 2, eight SMA dampers are placed at the bottom of the tower body. Four dampers are installed in the in-plane direction with two dampers on the first floor and the other two on the second floor. Similarly, four dampers are installed in the out-of-plane direction with two dampers on the first floor and the other two on the second floor. For scheme 3, eight SMA dampers are evenly installed in the middle of the tower body from the third floor to the sixth floor, as shown in Figure 6. 

The performance comparison of different control schemes is displayed in Figure 7. The structural peak responses are reduced substantially due to the installation of SMA dampers. The control performance of scheme 1 is slightly better than that of scheme 2. The control performance of scheme 3 is much worse than that of the other two schemes. Regarding scheme 2, eight SMA dampers are incorporated at the bottom of the tower body. The displacement responses of the tower bottom are less than those on top of the tower body. Relative small floor drifts at the tower bottom and small deformation of SMA can be observed. Thus, the energy dissipated by dampers is limited and the control efficacy is unsatisfactory. The overall control efficacy of scheme 1 is the best one, which is adopted in parametric studies and energy computation. The time histories of dynamic responses with and without SMA dampers are displayed in Figure 8. The controlled responses are much less than those of the original tower. The structural wind-induced responses are substantially suppressed for both two horizontal directions. The control efficacy of velocity is better than that of displacement and acceleration. The control performance of acceleration is slightly worse than that of displacement. 

The power spectral density (PSD) curves of dynamic responses of the controlled transmission tower are also plotted in Figure 9. The PSD curves of the fundamental vibrant mode are much larger compared with the other modes. This means that the major contribution of dynamic responses of a flexible truss tower is the first vibration mode. The peak PSD values of the controlled tower are much smaller than those of the uncontrolled tower. In addition, it is observed that the properties of PSD curves for the out-of-plane vibration are quite similar to the counterpart in the in-plane direction, which means that the control efficacy of SMA dampers for two horizontal directions is close. Thus, from the viewpoint of the frequency domain, the wind-excited responses of the structural system can be substantially suppressed by SMA dampers.

## 6. Parametric Study on Control Efficacy

### 6.1. Effect of Damper Stiffness

The stiffness coefficient (SC) of an SMA damper is defined as:(21)$$SC=\frac{k_{d}^{SMA}}{k_{0}^{SMA}}$$
where $k_{0}^{SMA}$ is the initial damper stiffness; $k_{d}^{SMA}$ is the stiffness in the parametric study. 

The influences of the damper stiffness on the structural peak responses are displayed in Figure 10. In the in-plane direction, the peak displacement gradually reduces with the increasing damper SC values until it increases to about 1.0. However, a further increment in SC cannot generate further significant displacement reduction. The varying trends of the peak velocity and acceleration are similar, as shown in Figure 10b,c. The optimal SC values for the peak velocity and acceleration are 2.0 and 1.0, respectively. Therefore, optimal SC values for various responses are different to some extent. Thus, it is not beneficial to accept a large stiffness coefficient to save fabrication costs. Similar observations are made in the out-of-plane direction, as shown in Figure 10d–f. In the out-of-plane direction of the structural system, the optimum SC values for the peak displacement, velocity, and acceleration are 1.0, 1.0, and 0.8, respectively. Therefore, the optimum SC value is selected as 1.0 considering the overall damper performance.

The variations in damper force and deformation with damper stiffness are investigated and displayed in Figure 11 and Table 2. The peak forces of SMA dampers are proportional to the SC values for both two directions. The peak forces of SMA dampers in the in-plane direction are larger than those in the out-of-plane vibration. Similarly, the peak deformation of SMA dampers is also examined and shown in Figure 11c,d and Table 2 for the two horizontal directions. The peak damper deformation gradually reduces with the increasing SC values. With the increasing SC value, the relative variations in peak forces are much larger than those of peak deformation. A very large damper force is disadvantageous for the damper movement and energy dissipation, which makes the damper behave as a steel brace. 

### 6.2. Influence of Damper Service Temperature

The variations in structural peak responses with service temperature are displayed in Figure 12. It is observed that the influences on peak responses of the in-plane vibration are relatively slight in comparison with those of damper stiffness, as displayed in Figure 12a–c. An optimal service temperature for the peak displacement and velocity of the tower top and cross-arm may exist. However, the peak responses of the tower body keep stable with varying service temperature. A similar observation can be made from the peak responses of the structural system in the out-of-plane direction. 

The variation trends in damper force and deformation are also examined and displayed in Figure 13 and Table 3. Similar to the effects of damper stiffness, the peak forces are proportional to the service temperature and the peak damper forces for the in-plane vibration are larger than those in the out-of-plane direction. However, the varying trend of the peak damper deformation is quite different from that of damper stiffness, as shown in Figure 13c,d and Table 3. With the increase in service temperature, the peak deformation of SMA dampers keeps stable for the in-plane movement. The peak damper deformation in the out-of-plane direction slightly increases with the increasing service temperature. Thus, the influence of service temperature on peak damper deformation is much smaller compared with that of damper stiffness. 

### 6.3. Variation of Hysteresis Loop

Configuration of hysteresis loops can reflect the control performance of a damper under wind excitations. Displayed in Figure 14 are the variations in hysteresis loops with damper stiffness for the two orthogonal directions. If a small stiffness is adopted (SC = 0.2), the SMA damper is easy to move and a large deformation is expected, as shown in Figure 14a. The enclosed area of the hysteresis loop is very small, which reflects a poor energy-dissipating capacity. To increase the damper stiffness (SC = 1.0), the shape of the hysteresis loops can be changed to a great extent, as displayed in Figure 14b. The damper deformation is reduced and the damper force is remarkably improved. In addition, the enclosed area increases remarkably and the control performance is substantially improved. If the damper stiffness continues to increase (SC = 3.0), the damper force can increase accordingly while the deformation reduces, as shown in Figure 14c. In this circumstance, the enclosed area does not increase and the energy-dissipating ability cannot be improved. For the out-of-plane vibration, the same conclusion can be drawn, as shown in Figure 14d–f. Thus, an optimal SC value can also be selected in line with the shapes of hysteresis loops. 

The variations in hysteresis loops with service temperature are also investigated through a detailed parametric study, as shown in Figure 15. Similar to the conclusions made from Figure 13, the peak damper forces quickly increase with the increasing service temperature for both two directions. If the service temperature is common (T = 0 °C), a relatively large damper deformation is observed and the hysteresis loop is plump which means satisfactory energy-dissipating capacity (See Figure 15a). If the service temperature gradually increases, the peak damper force also increases and the enclosed areas of SMA dampers reduce. If the service temperature reaches a relatively large value (T = 40 °C), the peak damper force increases quickly and, at the same time, the enclosed areas of SMA dampers dramatically reduce to a very small level, as shown in Figure 15c. In this circumstance, the SMA damper behaves like a steel brace. The peak displacement is reduced while the peak acceleration increases. This is due to the large peak damper force instead of a poor energy-dissipating capacity. Similar effects are observed from the hysteresis loops of SMA dampers for the out-of-plane vibration. Overall, the service temperature has a great influence on damper force instead of damper deformation. A very large service temperature is unnecessary for the improvement of the control performance of SMA dampers. 

It is noted that the real application of various energy-dissipating dampers in civil engineering structures will depend on the damper configuration and cost. For the transmission tower with SMA dampers, the amount of alloy used and the cost of SMA dampers are crucial issues that should be taken into consideration. From the viewpoint of real application, satisfactory control efficacy with optimal damper stiffness is essential. Optimal damper stiffness can be determined through parametric studies. A very large damper force is disadvantageous for the damper movement and energy dissipation, which makes the damper behave as a steel brace. Thus, it is not beneficial to accept a large stiffness coefficient to save fabrication costs. 

## 7. Properties of System Energy Responses

### 7.1. Energy Curves with Control

The control performance of SMA dampers on the structural system can also be illustrated by energy responses, as shown in Figure 16. For the uncontrolled transmission tower, the total input energy from wind loading *E_W_* is the sum of the kinetic energy *E_K_*, the strain energy *E_S_*, and structural damping energy *E_D_*, as shown in Figure 16a,b. Large *E_K_* and *E_S_* are observed due to the strong vibration of the entire coupled system. The kinetic and strain energy can only be dissipated by structural damping. 

The case for the controlled transmission tower is quite different, as displayed in Figure 16c,d. The magnitude of the kinetic energy *E_K_* and the strain energy *E_S_* are remarkably mitigated because the total sum of dissipated energy is substantially increased. The vibrant energy can be absorbed simultaneously by both the structural damping and SMA dampers. Owing to the contribution of SMA dampers, the structural damping energy *E_D_* is dramatically reduced. When comparing the energy curves without/with SMA dampers, the inputted energy *E_W_* is smaller compared with that of the uncontrolled tower. This is because the inputted energy is directly related to structural vibrant intensity. The dynamic responses of the controlled system are much smaller than those of the uncontrolled system. Thus, the inputted energy from wind excitations to the controlled tower *E_W_* is much smaller than that of the original tower. 

### 7.2. Effect of Damper Stiffness on Energy Response

The effects of damper parameters on structural energy responses are investigated through a parametric study in detail. The variations in energy responses with damper stiffness are examined and displayed in Figure 17. If damper stiffness is too small (SC = 0.2), the damper capacity in energy-dissipating is limited. As displayed in Figure 17a, the energy dissipated by SMA dampers *E_C_* quickly increases until the SC value reaches about 0.6. After that value, the *E_C_* gradually reduces. The energy *E_D_* is much smaller than that of the original tower, as shown in Figure 17b. The *E_D_* quickly reduces with the increase in SC values. It is also seen that the *E_D_* for SC = 1.0 is quite close to that for SC = 3.0, which means that a very large damper stiffness is unnecessary for the improvement of damper performance. The variations in total energy input from wind excitations *E_W_* with damper stiffness are also investigated and plotted in Figure 17c. Similar to the observations made from structural damping, firstly, the *E_W_* quickly reduces with the increasing SC values until SC reaches about 1.0. Then, a further increment in SC value cannot remarkably reduce the dynamic responses and the inputted energy *E_W_*. Furthermore, optimal damper SC values can be selected using the energy curves. 

The variations in energy responses with damper stiffness for the out-of-plane vibration are also investigated, as displayed in Figure 17d–f and similar conclusions to the in-plane vibration can be drawn. It is noted that the optimal SC values for different energy responses may differ to some extent. Furthermore, optimal SC values for the two directions are slightly different due to the difference in structural dynamic responses. Overall, the optimal SC value of SMA dampers for the example system can be selected as 1.0 in line with the energy responses, which is the same as that based on the peak responses, as shown in Figure 10.

### 7.3. Variation in Energy Response with Service Temperature

Figure 18 displays the variation in energy responses with service temperature. As shown in Figure 18a,b, with the increase in the service temperature, the energy dissipated by SMA dampers *E_P_* decreases while the energy dissipated by structural damping *E_D_* increases gradually. If the service temperature is common (T = 0 °C), the hysteresis loop is plump which means satisfactory energy-dissipating capacity, as shown in Figure 15a. If the service temperature is relatively large (T = 40 °C), the peak damper force increases quickly and the SMA damper behaves like a steel brace, as shown in Figure 15c. The enclosed areas of the hysteresis loop dramatically decrease and the energy dissipated by SMA dampers *E_P_* reduces, as shown in Figure 18a. 

The variations in the inputted energy *E_W_* with service temperature are shown in Figure 18c, which are different from those with damper stiffness, as displayed in Figure 17c. The increment in the service temperature cannot improve the *E_C_* of SMA dampers but increase the damper force, as displayed in Figure 15. Thus, the effects of service temperature on structural peak responses are small (see Figure 12). As mentioned above, the inputted energy is directly related to structural vibrant intensity. The influence of service temperature on the inputted energy *E_W_* is relatively small. The energy curves for the out-of-plane vibration present similar results, as shown in Figure 18d–f. On the whole, a very large service temperature is not beneficial for the performance of SMA dampers. An optimal service temperature of SMA dampers for the example structural system can be selected as T = 0 °C based on the energy curves, which is the same as that based on the peak responses, as shown in Figure 12. 

## 8. Concluding Remarks

The vibration control of a tower-line coupled system disturbed by wind loading was conducted by SMA dampers. The analytical model of the entire system was established based on Lagrange’s equations by considering the dynamic interaction between transmission lines and towers. The control efficacy was analyzed in both the time domain and the frequency domain. Detailed parametric studies were conducted to examine the influence of damper stiffness, service temperature on structural responses, and hysteresis loops. The following conclusions can be drawn:
(1)SMA dampers are beneficial in the vibration control of the tower-line coupled system disturbed by wind loading. The control efficacy on displacement and velocity is slightly better than acceleration. The peak PSD values of the controlled tower are much smaller than those of the uncontrolled tower. (2)The peak responses gradually decrease with the increasing damper stiffness. For the in-plane vibration, the optimal SC values for the peak displacement, velocity, and acceleration are 1.0, 2.0, and 1.0, respectively. For the out-of-plane vibration, the optimal SC values for the peak displacement, velocity, and acceleration are 1.0, 1.0, and 0.8, respectively. Therefore, the optimum SC value is selected as 1.0 considering the overall damper performance. An optimum stiffness coefficient exists for the response control and it is unnecessary to set a very large stiffness coefficient to save fabrication cost. (3)The influence of service temperature on peak damper deformation is much smaller compared with that of damper stiffness. If the service temperature reaches a relatively large value (T = 40 °C), the peak damper force increases quickly and, at the same time, the enclosed areas of SMA dampers dramatically reduce to a very small level. Overall, the service temperature has a great influence on damper force instead of damper deformation. A very large service temperature is unnecessary for the improvement of the control performance of SMA dampers.(4)The control performance on wind-induced dynamic responses can also be depicted by energy responses. Furthermore, the optimal stiffness coefficient and service temperature of SMA dampers can also be determined in line with energy curves.

## Figures and Tables

**Figure 1 materials-15-01790-f001:**
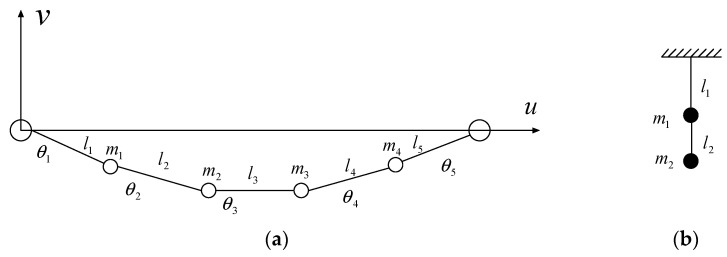
Analytical model of a transmission line. (**a**) In-plane vibration; (**b**) out-of-plane vibration.

**Figure 2 materials-15-01790-f002:**
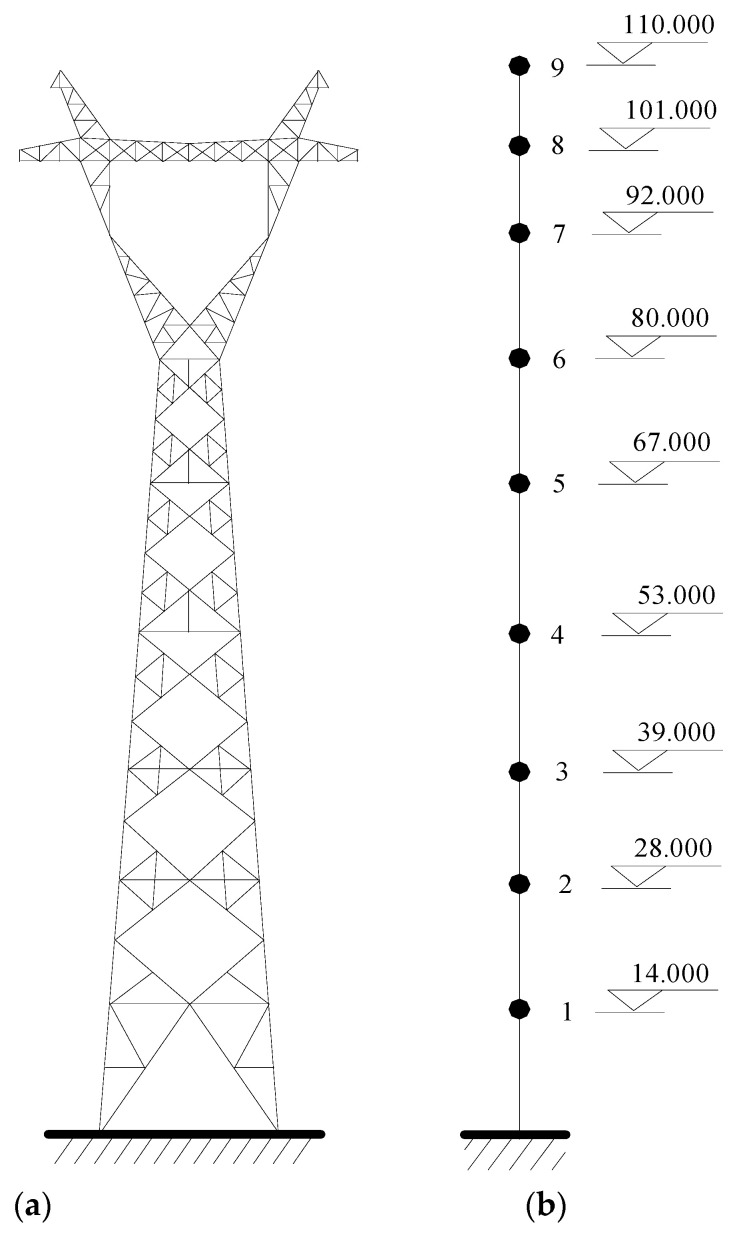
Analytical model of a large transmission tower. (**a**) 3D FE model; (**b**) 2D dynamic model.

**Figure 3 materials-15-01790-f003:**
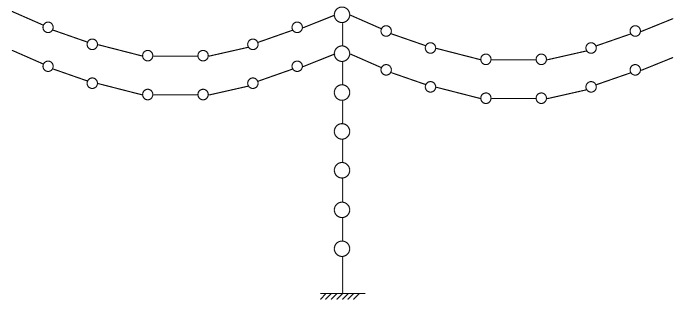
Analytical model of a transmission tower-line system.

**Figure 4 materials-15-01790-f004:**
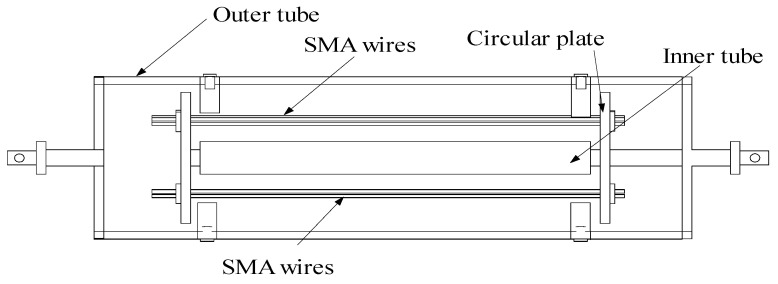
Configuration of an SMA damper.

**Figure 5 materials-15-01790-f005:**
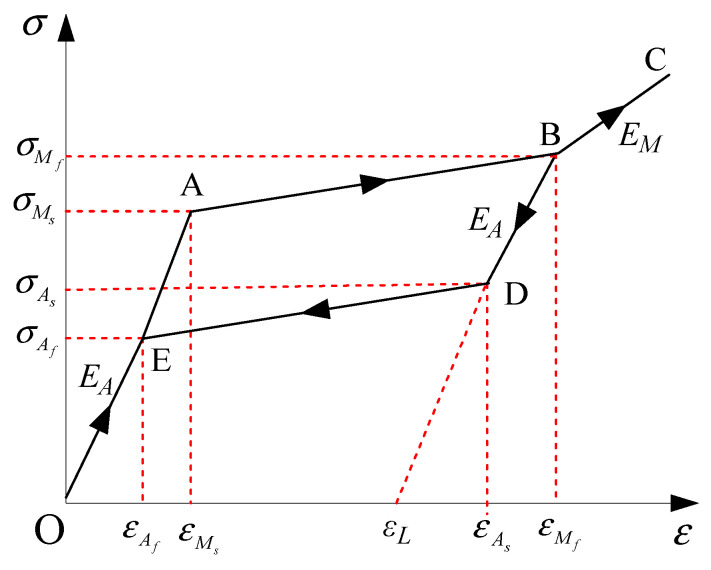
Hysteretic model of an SMA wire.

**Figure 6 materials-15-01790-f006:**
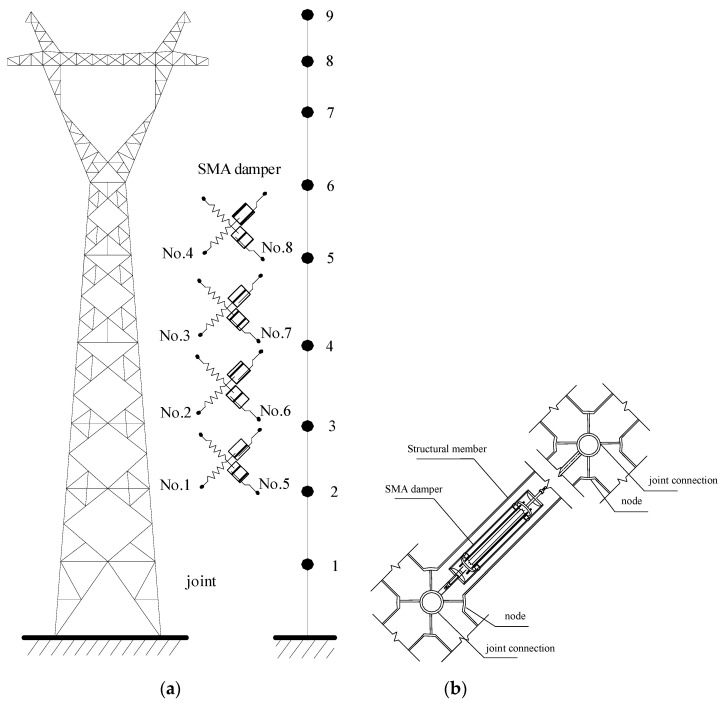
Installation scheme of SMA dampers. (**a**) Damper location; (**b**) Damper connection.

**Figure 7 materials-15-01790-f007:**
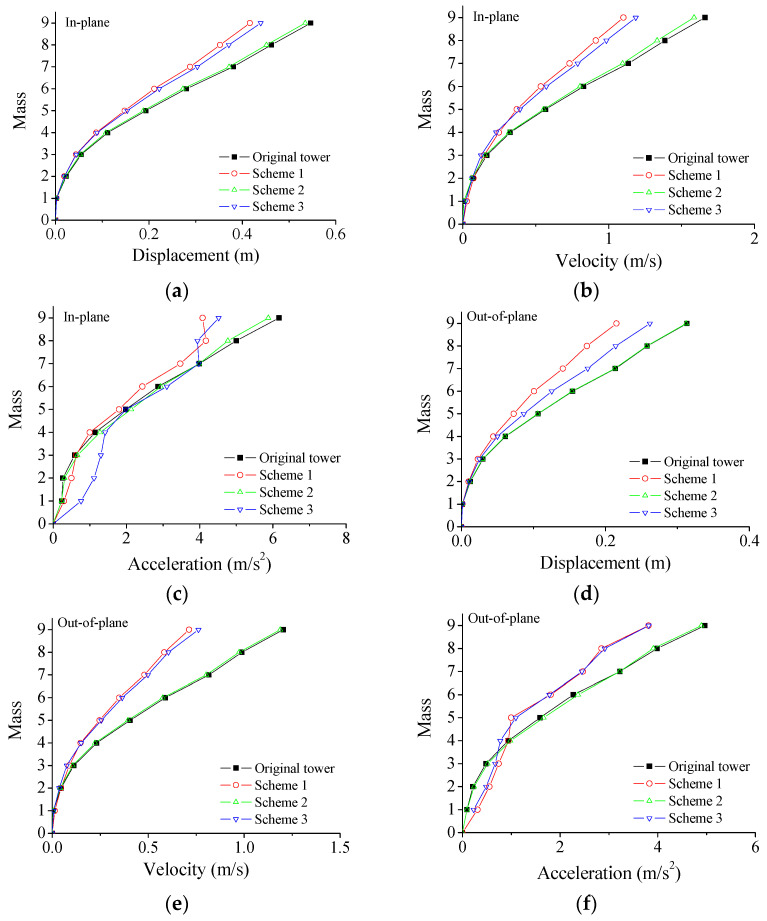
Comparison of different control schemes. (**a**) Peak displacement; (**b**) peak velocity; (**c**) peak acceleration; (**d**) peak displacement; (**e**) peak velocity; (**f**) peak acceleration.

**Figure 8 materials-15-01790-f008:**
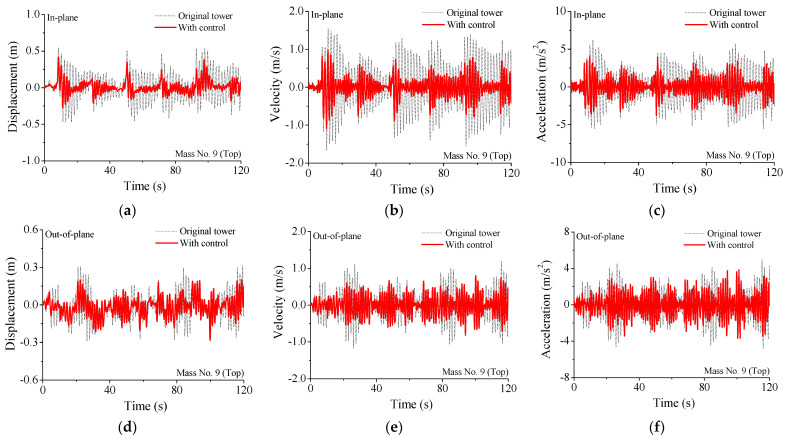
Time histories of dynamic responses of the transmission tower-line system. (**a**) Displacement response; (**b**) velocity response; (**c**) acceleration response; (**d**) displacement response; (**e**) velocity response; (**f**) acceleration response.

**Figure 9 materials-15-01790-f009:**
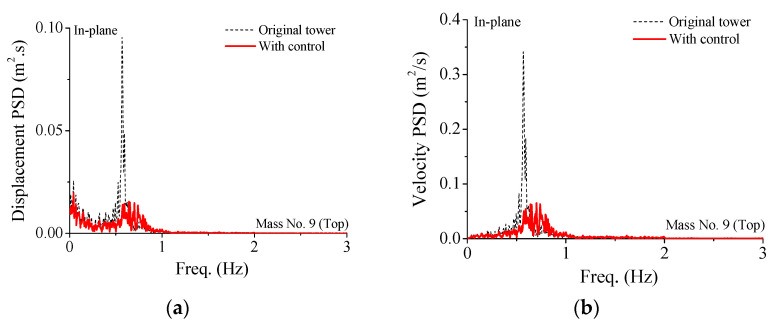
Comparison of PSD curves of the transmission tower-line system. (**a**) Displacement PSD curve; (**b**) velocity PSD curve; (**c**) acceleration PSD curve; (**d**) displacement PSD curve; (**e**) velocity PSD curve; (**f**) acceleration PSD curve.

**Figure 10 materials-15-01790-f010:**
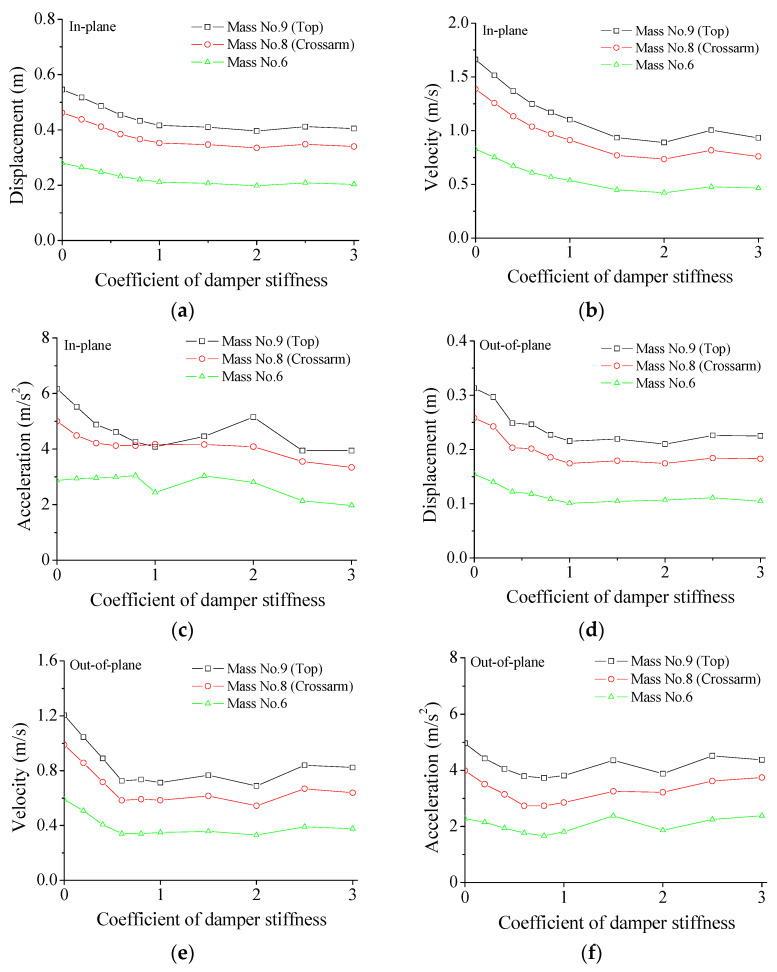
Stiffness effects of SMA damper on peak responses. (**a**) Peak displacement; (**b**) peak velocity; (**c**) peak acceleration; (**d**) peak displacement; (**e**) peak velocity; (**f**) peak acceleration.

**Figure 11 materials-15-01790-f011:**
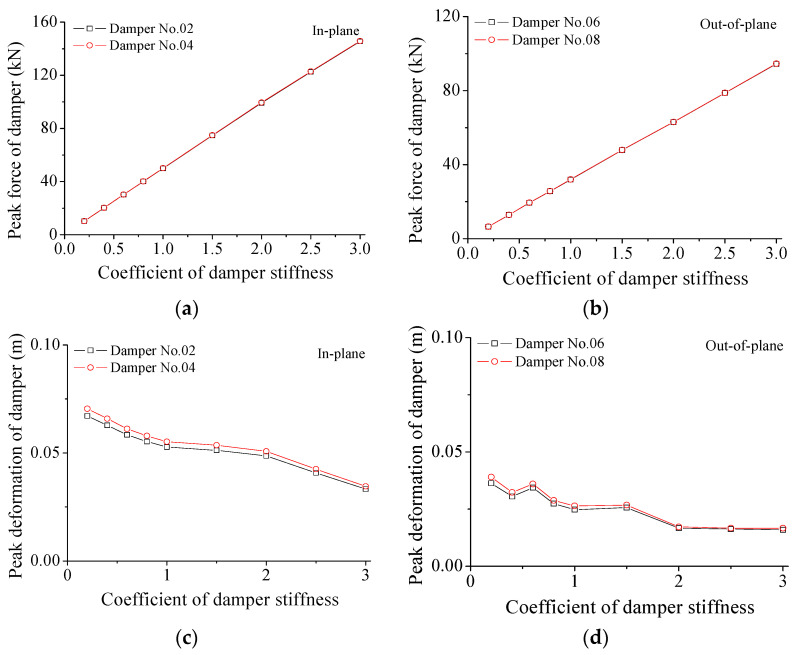
Variation in damper force and deformation with damper stiffness. (**a**) Damper force for the in-plane direction; (**b**) damper force for the out-of-plane direction; (**c**) damper deformation for the in-plane direction; (**d**) damper deformation for the out-of-plane direction.

**Figure 12 materials-15-01790-f012:**
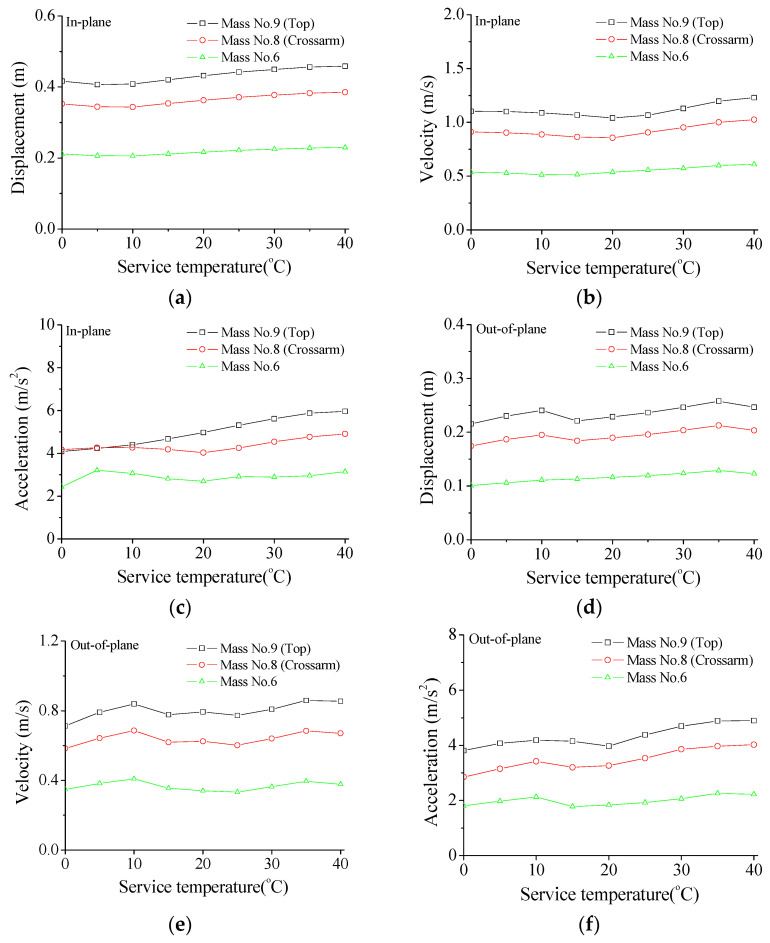
Effects of service temperature on maximum responses of the transmission tower. (**a**) Peak displacement; (**b**) peak velocity; (**c**) peak acceleration; (**d**) peak displacement; (**e**) peak velocity; (**f**) peak acceleration.

**Figure 13 materials-15-01790-f013:**
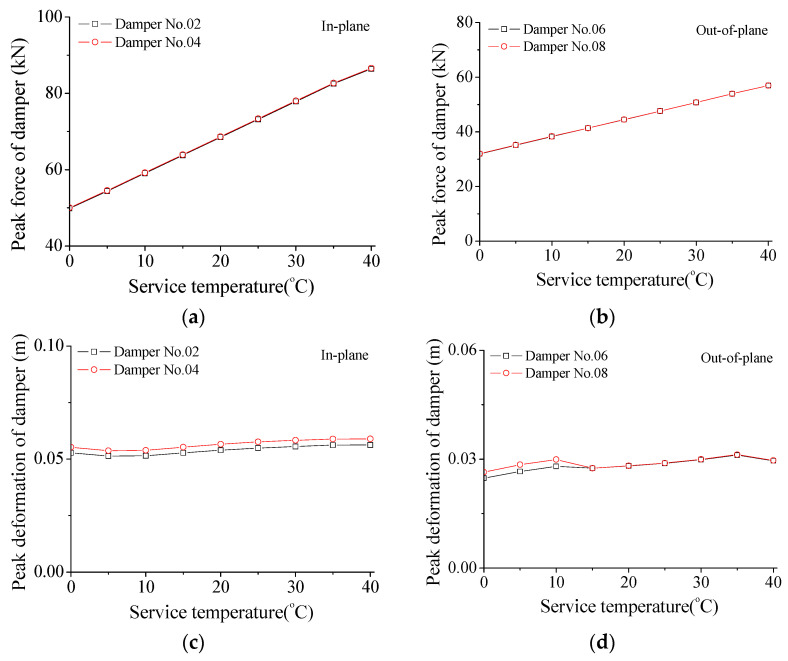
Variation in damper force and deformation with service temperature. (**a**) Damper force for the in-plane direction; (**b**) damper force for the out-of-plane direction; (**c**) damper deformation for the in-plane direction; (**d**) damper deformation for the out-of-plane direction.

**Figure 14 materials-15-01790-f014:**
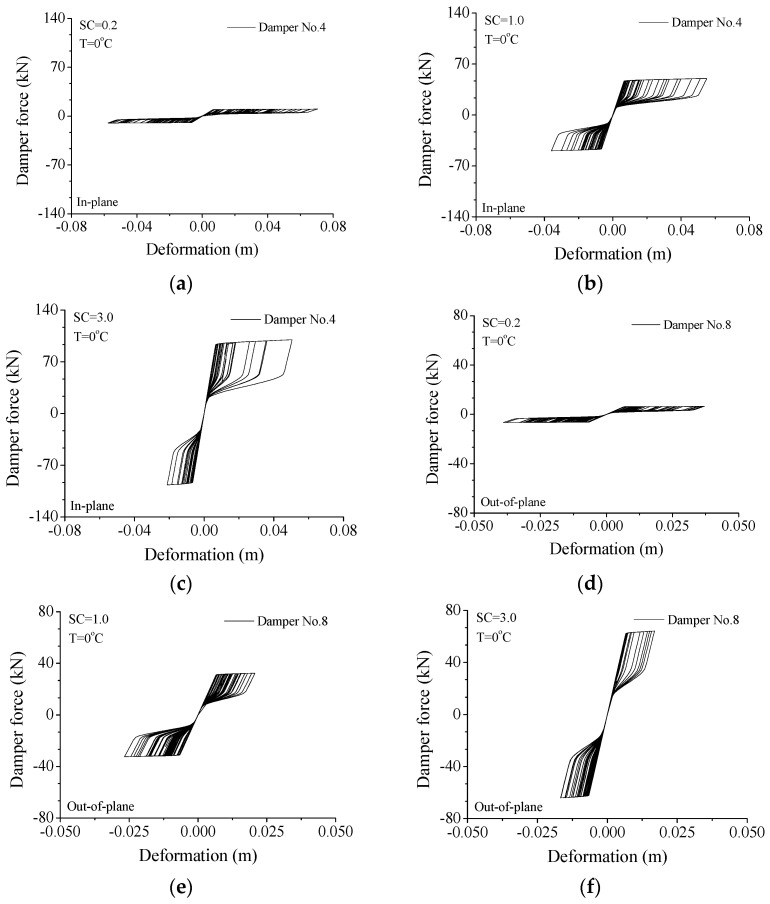
Variation in hysteresis loop with damper stiffness. (**a**) SC = 0.2; (**b**) SC = 1.0; (**c**) SC = 3.0; (**d**) SC = 0.2; (**e**) SC = 1.0; (**f**) SC = 3.0.

**Figure 15 materials-15-01790-f015:**
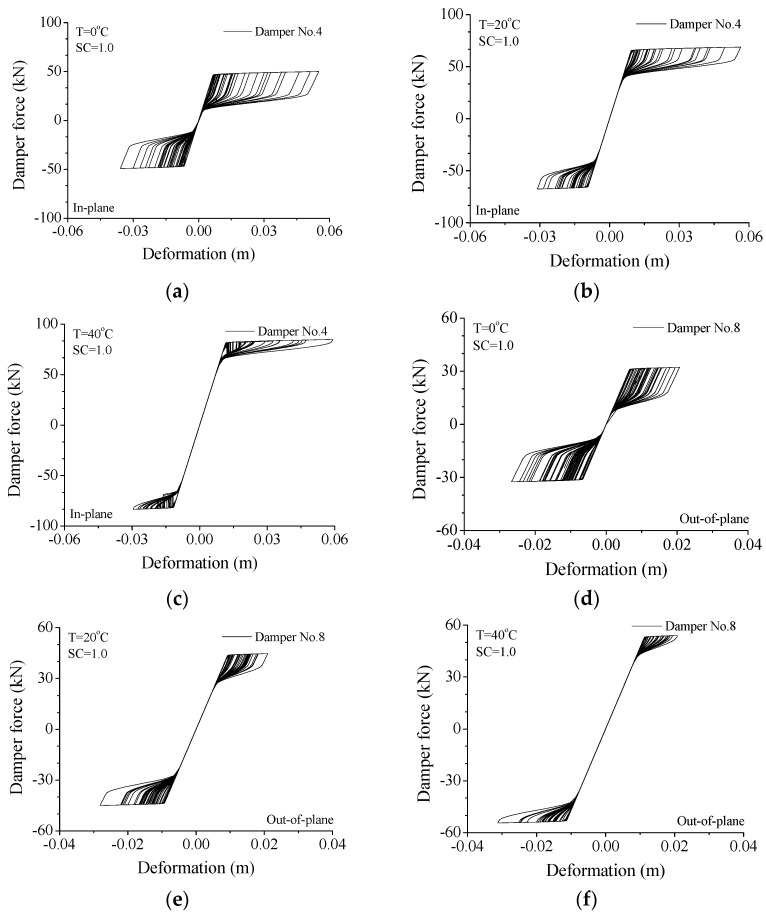
Variation in hysteresis loop with service temperature. (**a**) T = 0 °C; (**b**) T = 20 °C; (**c**) T = 40 °C; (**d**) T = 0 °C; (**e**) T = 20 °C; (**f**) T = 40 °C.

**Figure 16 materials-15-01790-f016:**
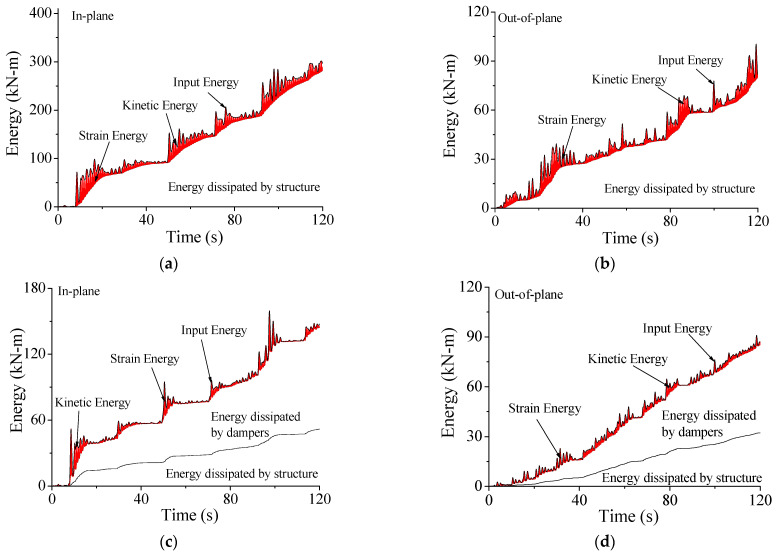
Energy responses of the transmission tower without/with SMA dampers. (**a**) Tower energy for in-plane vibration; (**b**) tower energy for out-of-plane vibration; (**c**) tower energy with control for in-plane vibration; (**d**) tower energy with control for out-of-plane vibration.

**Figure 17 materials-15-01790-f017:**
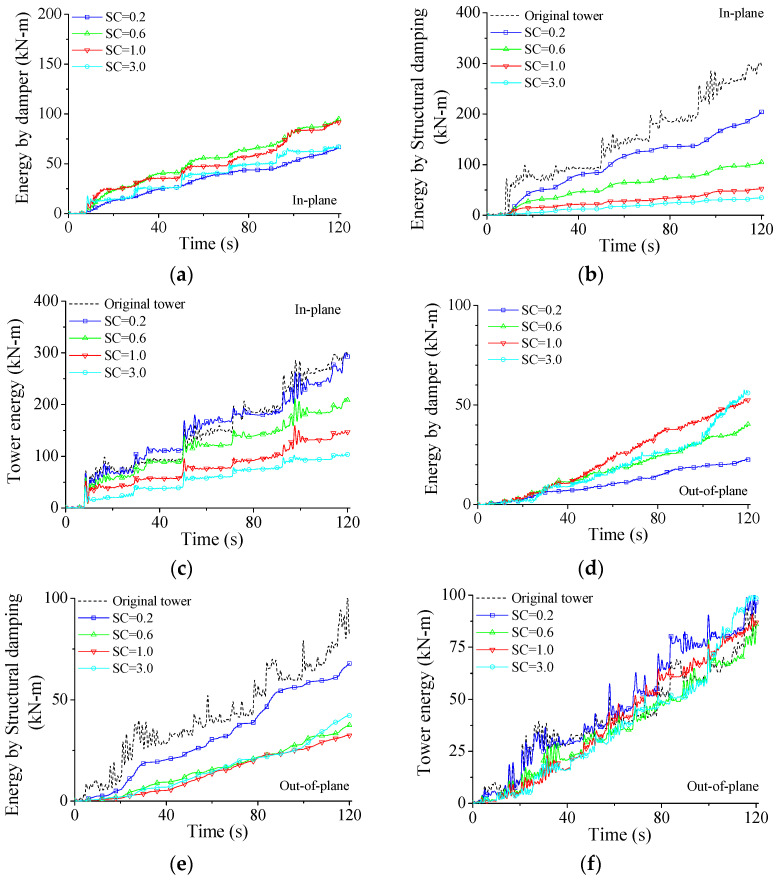
Variation in energy responses with damper stiffness. (**a**) Tower energy for in-plane vibration; (**b**) structural damping energy for in-plane vibration; (**c**) damper energy for in-plane vibration; (**d**) tower energy for out-of-plane vibration; (**e**) structural damping energy for out-of-plane vibration; (**f**) damper energy for out-of-plane vibration.

**Figure 18 materials-15-01790-f018:**
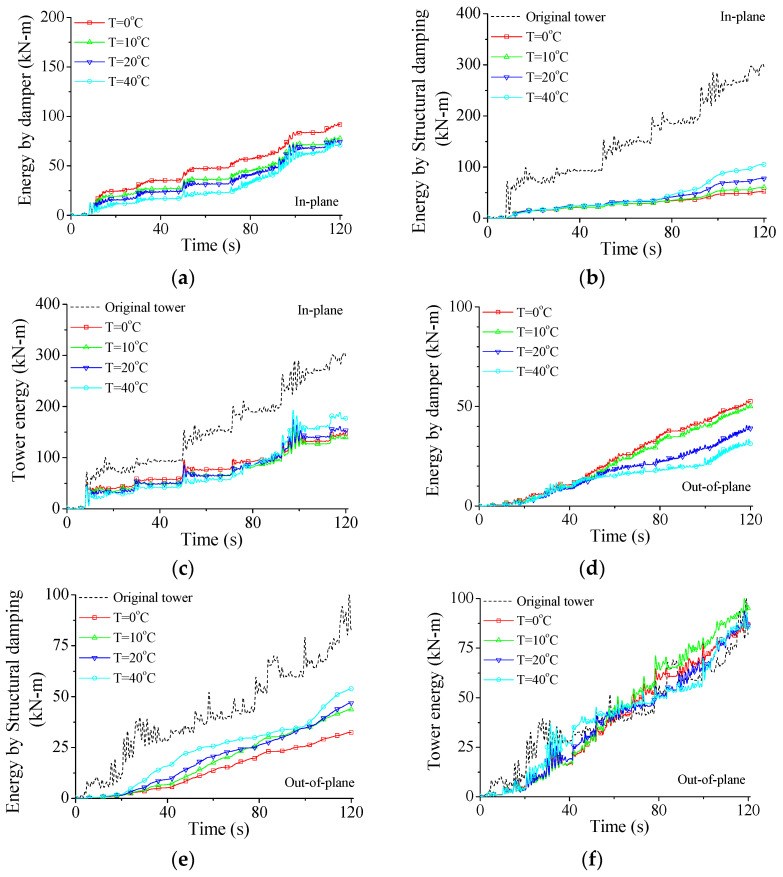
Variation in energy responses with service temperature. (**a**) Tower energy for in-plane vibration; (**b**) structural damping energy for in-plane vibration; (**c**) damper energy for in-plane vibration; (**d**) tower energy for out-of-plane vibration; (**e**) structural damping energy for out-of-plane vibration; (**f**) damper energy for out-of-plane vibration.

**Table 1 materials-15-01790-t001:** Chemical composition of Q235 steel.

A	C ≤ 0.22%	Mn ≤ 1.4%	Si ≤ 0.35%	S ≤ 0.050%	*p* ≤ 0.045%
B	C ≤ 0.20%	Mn ≤ 1.4%	Si ≤ 0.35%	S ≤ 0.045%	*p* ≤ 0.045%
C	C ≤ 0.17%	Mn ≤ 1.4%	Si ≤ 0.35%	S ≤ 0.040%	*p* ≤ 0.040%
D	C ≤ 0.17%	Mn ≤ 1.4%	Si ≤ 0.35%	S ≤ 0.035%	*p* ≤ 0.035%

**Table 2 materials-15-01790-t002:** Variations in peak force and deformation of SMA dampers with damper stiffness.

Damper. No.		SC = 0.4	SC = 1.0	SC = 2.0	SC = 3.0
02 In-plane direction	Peak force	20.24 kN	49.86 kN	99.16 kN	145.44 kN
	Peak defomation	6.28 cm	5.27 cm	4.86 cm	3.33 cm
06 Out-of-plane direction	Peak force	12.87 kN	31.91 kN	63.04 kN	94.47 kN
	Peak defomation	3.05 cm	2.48 cm	1.66 cm	1.61 cm

**Table 3 materials-15-01790-t003:** Variations in peak force and deformation of SMA dampers with service temperature.

Damper. No.		T = 0 °C	T = 10 °C	T = 20 °C	T = 40 °C
02 In-plane direction	Peak force	49.86 kN	59.04 kN	68.48 kN	86.39 kN
	Peak defomation	5.28 cm	5.14 cm	5.39 cm	5.62 cm
06 Out-of-plane direction	Peak force	31.91 kN	38.26 kN	44.47 kN	56.94 kN
	Peak defomation	2.49 cm	2.81 cm	2.82 cm	2.96 cm

## Data Availability

Date sharing is not applicable to this article.

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
