# Peer review of "Vibration Control of a Wind-Excited Transmission Tower-Line System by Shape Memory Alloy Dampers"

_materials, 2022, doi:10.3390/ma15051790_

Round 1
Reviewer 1 Report
I kindly ask you to read the attached file, and please do all the chnages. Very good ideas.

Reviewer 2 Report
It is advisable to give a brief review of the literature in the introduction. For example, it is written that "Chen et al [17] examined the seismic responses of a 340m television tower reinforced using friction dampers", but it is unclear what results the author received in source No. 17. It would be nice to reflect this very briefly in 1-2 sentences after the links. Not only for reference #17, but also #18 and so on.
At the end of page 2 there is a text: "Where ξi (i=2,3,4) and δi (i=1,2,..,5)». You write that it is a «structural generalized coordinates related to the θ and l". But θ and l changed from 1 to 5. And in your article ξi changed from 2 to 3, and in the equation (2) from 2 to 4. It is necessary to lead to a single picture or explain why this is so.
It would be good to expand Figure 6 and show all 3 air-cushions arrangement schemes on it.
In Figure 7, "Mass" is a dimensionless quantity, why is the dimension not specified? Although there are values for it.
Under Figure 10 it is written: "However, a further increase in SC cannot generate further significant displacement reduction." It would be good to add why there is no further increase. Also in the same description, give an explanation of why exactly SC optimal values for the peak velocity market and acceleration are 2.0 and 1.0.
At the beginning of section 5.1, you write that the supports are made of Q235 steel. It would be nice to show the chemical composition of this brand in the table or in the text and write what type of steel this is. This will help bring the article closer to the topic of the materials and it will more correspond to the subject of the journal. Now the article mainly provides data on the study of the design. In my opinion, this is closer to the journal "Applied Science".
The conclusions should be expanded. A lot of work has been done in the article. The conclusions present only general results without numerical data. It would be good to cite the most important numerical results of the calculations.

Reviewer 3 Report
The authors have investigated the vibration control using SMA dampers for the tower-line coupled system. The control efficacy is verified in both the time domain and the frequency domain. Detailed parametric studies are conducted to examine the effects of physical parameters of SMA dampers on structural responses and hysteresis loops. In addition, the structural energy responses are computed to examine the control performance.
The manuscript is well-written and fits within the journal's scope. However, before a final decision can be made, the following comments should be responded to.
- The authors should clearly highlight the novelty of the work.
- Does the configuration of the SAM dampers used, is proposed by the authors?
- Figure 4: Configuration of an SMA damper should be presented in 3D space for a better presentation.
- In Figure 8, it seems that the Displacement response has been better controlled compared to Velocity and Acceleration. A clarification is needed.
- The developed MATLAB program should be added as an Appendix or as a supplementary source.
- The conclusion section should be re-written in bullet point format. The main contributions should be given in terms of values or percentages.
- The Literature review can be extended including the following references:
- Han, Y. L., Li, Q. S., Li, A. Q., Leung, A. Y. T., & Lin, P. H. (2003). Structural vibration control by shape memory alloy damper. Earthquake engineering & structural dynamics, 32(3), 483-494.
- Tabrizikahou, A., Kuczma, M., Łasecka-Plura, M., & Noroozinejad Farsangi, E. (2022). Cyclic Behavior of Masonry Shear Walls Retrofitted with Engineered Cementitious Composite and Pseudoelastic Shape Memory Alloy. Sensors, 22(2), 511.
- Li, X. H., & Zhu, Z. W. (2022). Nonlinear dynamic characteristics and stability analysis of energy storage flywheel rotor with shape memory alloy damper. Journal of Energy Storage, 45, 103392.
Round 2
Reviewer 2 Report
According to comments 6 and 3, it was recommended to include a table/figure in the text of the article. I didn't ask you to explain to me. I asked to clarify this in the text of the article for readers. You either provide data in the article or write why you do not agree.
The rest of the remarks have been corrected, thanks to the authors.
Reviewer 3 Report
The authors have fully implemented the reviewer's comments.
